# Diversity and Correlation Analysis of Endophytes and Top Metabolites in *Phlomoides rotata* Roots from High-Altitude Habitats

**DOI:** 10.3390/microorganisms13030503

**Published:** 2025-02-25

**Authors:** Zuxia Li, Huichun Xie, Guigong Geng, Chongxin Yin, Xiaozhuo Wu, Jianxia Ma, Rui Su, Zirui Wang, Feng Qiao

**Affiliations:** 1Key Laboratory of Tibetan Plateau Medicinal Plant and Animal Resources, School of Life Sciences, Qinghai Normal University, Xining 810008, China; 18909789774@163.com (Z.L.); yinchongxin0701@163.com (C.Y.); xiaozhuo0623@163.com (X.W.); majianxia0926@163.com (J.M.); 18797079134@163.com (R.S.); wangzirui020124@163.com (Z.W.); 2Academy of Plateau Science and Sustainability, Qinghai Normal University, Xining 810008, China; 3Qinghai South of Qilian Mountain Forest Ecosystem Observation and Research Station, Huzhu 810500, China; genggg-298@163.com; 4Academy of Agricultural and Forestry Sciences, Qinghai University, Xining 810016, China

**Keywords:** *Phlomoides rotata*, endophytes, soil physicochemical indicators, correlation analysis

## Abstract

*Phlomoides rotata*, a traditional medicinal plant, always grows on the Tibetan Plateau at a high altitude of 3100–5200 m. The major active ingredients in *P. rotata* were used in medicines due to their diverse pharmacological effects, including hemostatic, anti-inflammatory, antitumor, immuno-modulatory, and antioxidant activities. This study screened 15 top endophytic genus through the analysis of OTUs and the top 30 metabolites with relatively high content in *P. rotata* roots from four different habitats (HN, GL, YS, and CD regions) in Qinghai Province. Twelve physicochemical indicators were measured and analyzed in the rhizosphere soils of *P. rotata* habitats. The results indicated that the top 30 metabolites compounds included 7 amino acids, 5 sugars and alcohols, 4 phenylpropanoids, 3 Organic acids, and 3 Alkaloids. Four endophytic bacteria (*Acidibacter*, *Sphingomonas, Variovorax,* and *Sphingobium*) and three endophytic fungi (*Tetracladium, Cadophora,* and *Minimelanolocus*) were dominant genera in *P. rotata* roots from four habitats. There were 109 positive significant correlations and 57 negative correlations between OTUs of endophytic bacteria and contents of top 30 metabolites, and 59 positive significant correlations and 58 negative correlations between OTUs of endophytic fungus and contents of top 30 metabolites. The OTUs of *Acidibacter* were significantly positively correlated with the content of 5 soil physicochemical indicators (total phosphorus, amylase, sucrase, total potassium, or soil organic carbon) and significantly negatively correlated with the content of acid protease. OTUs of *Tetracladium* or *Cadophora* showed a positive correlation with the content of total phosphorus and a negative correlation with that of alkaline phosphatase. This study provides a theoretical basis for the study of the correlation between endophytes and metabolites in *P. rotata* roots.

## 1. Introduction

Endophytes are an important component of the plant microbiome and are typically found within the healthy tissues of living plants in the form of bacterial and fungal communities [1,2]. Highly diverse endophytic communities can considerably influence the metabolite composition of host plants [3]. Through coevolution, some endophytic bacteria and their host plants have established specialized relationships that can significantly influence the formation of metabolites within a plant, affecting the quality and quantity of medicinal plant raw materials [4]. Endophytic microorganisms absorb nutrients and prevent pathogen damage, supporting healthy plant growth [4]. Liu et al. (2020) investigated the relationship between endophytic fungi and berberine synthesis in the medicinal plant *Coptis teeta* Wall [5]. Proteobacteria, Actinobacteria, and Bacteroidetes were the major phyla, while Mycobacterium, Salmonella, Nocardioides, Burkholderia, Paraburkholderia, and Rhizobium were the dominant genera in the root, stem, and leaf tissues of *C. teeta* [5]. Their results demonstrated a clear correlation between dominant endophytic fungi and berberine synthesis in *C. teeta* [5]. These studies have emphasized the crucial role of diverse bacterial or fungi communities in various plant organs in maintaining plant health and growth. In addition, endophytes have been shown to be closely related to the accumulation of differential metabolites [6,7]. A study on the phenolic acid content of the male and female leaves of *Alsophila spinulosa* and the diversity of endophytic bacteria revealed a close correlation between the abundance of endophytic mycobiome and differences in metabolite composition [8]. Furthermore, research on the correlation between metabolomics and endophytic fungal communities in different desert regions of China identified 13 distinct labeled metabolites and 7 shared fungal operational taxonomic units (OTUs), indicating a significant correlation between metabolites and endophytic fungi [9]. These findings suggest that endophytic microorganisms, including bacteria and fungi, may be key factors that influence the composition of secondary metabolites in plants [10]. In summary, a close relationship exists between endophyte in plants and their environment and the types and contents of secondary metabolites [11].

The interaction between plants and microorganisms is crucial for the health and productivity of plants, mainly occurring in the rhizosphere: the narrow soil area around the roots of living plants [12]. The interaction between plants and the microbiome is essential for plant nutrition acquisition [13]. However, we know very little about the reactions of plant roots, clumps, and rhizosphere soil microbial communities [14]. The key component that affects the rhizosphere ecological factors of plants is the feedback mechanism between plants and the soil environment [15]. Study the composition of rhizosphere soil and root endophytic bacteria in *Cotoneaster acutifolius* Turcz., *Lonicera japonica* Thunb., and *Cornus alba* L. [16]. The results revealed that the dominant phyla of rhizosphere soil bacteria and root-endophytic bacteria in different shrubs were *Proteobacteria* and *Actinobacteria*. Redundancy analysis (RDA) and Pearson correlation analysis revealed that soil dehydrogenase, soil N-acetyl-β-D glucosidase, alkaline protease, pH, and total phosphorus were the main influencing factors on the bacterial community structures in root-endophytic bacteria (*p* < 0.05), while alkaline protease, pH, total carbon, and total nitrogen significantly impacted rhizosphere soil community structures (*p* < 0.05) [16]. Physicochemical parameters like nitrogen, phosphorus, and potassium were found to be significantly positively correlated with Muribaculaceae highly enriched in the non-saline zone [17,18]. In addition, soil pH, Total Nitrogen (TN), Available Phosphorus (AP), Substrate-Induced Respiration (SIR), conversion enzymes, and proteases are considered the main factors affecting the classification and functional group changes of rhizosphere and bulk soil microbial communities [19,20].

*Phlomoides rotata* (Benth. ex Hook. f.) Mathiesen is a perennial herb that grows in meadows, grassland, and gravel areas at an altitude of 3100–5100 m [21]. *P. rotata* is mostly distributed in Qinghai, Xizang, Sichuan, and Gansu Provinces, China, particularly on the Qinghai–Tibet Plateau [22]. *P. rotata* has been studied with regard to its traditional uses, chemical constituents, pharmacological effects, clinical applications, quality control, toxicology, and pharmacokinetics [23] About 127 chemical components have been isolated and identified from *P. rotata*, and they mostly include flavonoids, phenylethanol glycosides (PhGs), and iridoids [24]. These secondary metabolites exhibit a wide range of pharmacological properties, including hemostatic [25,26,27], anti-inflammatory [28], immunomodulatory [29], antioxidant activities [30], renal protection [31], and liver protective activity [32]. Here, *P. rotata* is native to the Qinghai–Tibet Plateau region, and its unique geographical environment results in its unique pharmacological activities.

Endophytes have elicited increasing attention due to their capacity to produce a vast array of bioactive secondary metabolites [33]. However, endophytic colonies in *P. rotata* roots from high-altitude habitats remain unclear and largely uncharacterized. Nevertheless, information regarding the relationship between endophytic diversity and secondary metabolism in *P. rotata* roots is scarce. In the current study, the composition and diversity of endophytes in *P. rotata* roots from four different habitats at an altitude of 3500–4300 m were explored using ITS2 and 16S ribosomal RNA sequencing techniques. Ultrahigh-performance LC (UPLC)–MS was performed to explore the distribution of metabolites in *P. rotata* roots from four different habitats. The Pearson statistical method was used to analyze the possible correlation between endophytes and major active metabolites in *P. rotata.* The current study is expected to lay a foundation for the further understanding of endophytes and secondary metabolites in *P. rotata*.

## 2. Materials and Methods

### 2.1. Plant Materials

During the flowering period, *P. rotata* roots were collected from four habitats in Qinghai Province, with altitudes from 3540 m to 4270 m (Table 1). The sites were Henan County (HN, 3540 m), Guoluo County (GL, 3750 m), Yushu County (YS, 3880 m), and Chengduo County (CD, 4270 m). Only healthy, similar-sized *P. rotata* plants were chosen. 10–15 healthy, uniformly growing plants were randomly selected from each area. The whole plants were carefully dug out, washed repeatedly to remove topsoil, sterilized with 75% ethanol for 5 min, and rinsed twice with sterile water. The roots, used as experimental materials, were immediately frozen in liquid nitrogen and then stored at −80 °C for metabolomic and microbiological analysis.

### 2.2. DNA Extraction and High-Throughput Sequencing of Endophytes in P. rotata Roots

The extracted sample DNA was used as a template, and the universal primer for the 16S rDNA V3-V4 region (F: CADACTCCTACGGGAGGC and R: ATCCTGTTTGMTMCCCVCRC) was applied. A sequencing adapter was added to the end of the primer, and PCR amplification was performed. The product was then purified, quantified, and homogenized to form a sequencing library. The constructed library was first subjected to library quality inspection, and qualified libraries were sequenced using Illumina NovaSeq 6000. For quality filtering, the Raw Reads obtained from sequencing were first filtered using Trimmatic v0.33 software. Then, Cutadapt 1.9.1 software was used to identify and remove primer sequences, obtaining Clean Reads without primer sequences. The dada2 [34] method from QIIME2 2020.6 [35] was used for denoising, concatenating the two end sequences, and removing chimeric sequences, resulting in the final valid data (Non-chimeric Reads).

### 2.3. Diversity Analysis of Endophyties in P. rotata Roots

The sequencing results were subjected to quality control and denoising using the QIIME2 2020.6 [35] plug-in to obtain valid data. According to the similarity, the sequence was clustered into the operational taxonomic unit (OTU), and species annotation was performed. The alpha diversity index was calculated using QIIME software (version 2020.6.0, https://qiime2.org, 11 January 2025). When using the Python Lefse (version 1.1.1, https://github.com/SegataLab/lefse/tree/master/lefse, 11 January 2025) package for LDA analysis, the first step was to use the nonparametric factor Kruskal Wallis (KW) sum rank tested to detect significant differences in abundance characteristics and identify groups with significant differences in abundance. Then, linear discriminant analysis (LDA) was used to estimate the magnitude of the impact of each component (species) abundance on the difference effect [36].

### 2.4. Extraction of Metabolites

Metabolites were extracted from *P. rotata* roots of different regions, with three biological replicates. Firstly, samples were vacuum freeze-dried, and 50 mg was weighed, then 1000 μL of extraction solution (methanol: acetonitrile: water = 2:1:1) was added. Next, the samples were ground (45 Hz, 10 min) and sonicated in an ice water bath for 10 min. After that, the samples were placed in a −20 °C freezer for 1 h. The static samples were centrifuged at 4 °C and 12,000 rpm for 10 min. The 500 μL supernatant was collected and dried by a vacuum concentrator, and 160 μL 50% acetonitrile solution was added to dissolve the dried extract. The dissolved samples were vortexed, put back in the ice water bath for 10 min, and centrifuged again (4 °C, 12,000 rpm, 10 min). Finally, the 120 μL supernatant was collected and stored in a 2 mL injection bottle for LC–MS/MS detection and analysis [37].

### 2.5. LC-MS/MS Conditions

LC–MS/MS analysis was performed using a Whatsch Acquisition I-Class PLUS ultrahigh-performance liquid chromatography-tandem AB Scienx Qtrap 6500+ high-sensitivity mass spectrometer. The conditions for ultrahigh-performance liquid chromatography were as follows: the chromatographic column was a Waters Acquisition UPLC HSS-T3 (1.8 µm, 2.1 mm × 100 mm). Mobile Phase A was ultrapure water (containing 0.1% formic acid and 5 mM ammonium acetate), and Phase B was acetonitrile (with 0.1% formic acid added). The gradient elution program was used for sample measurement, with initial conditions of 98% A and 2% B maintained for 1.5 min and then adjusted to 50% A and 50% B within 5 min. The linear gradient was adjusted to 2% A and 98% B within 9 min, maintained for 1 min, adjusted to 98% A and 2% B within 1 min, and maintained for 3 min. The flow rate was set to 0.35 mL/min, the column temperature was set to 50 °C, and the flow rate was 350 μL/min.

### 2.6. Soil Sample Collection and Physicochemical Property Measurements

Soil samples were collected from the rhizosphere of *P. rotata*. The collected soil samples undergo artificial homogenization treatment to ensure the removal of roots and coarse plant debris, thereby obtaining composite samples. Subsequently, store the composite sample in a well-ventilated and cool place for drying. Assess the physical and chemical properties of soil. According to Guan Song Yin’s” Soil Enzymes and Their Research Methods” [38], the activities of sucrase, amylase, polyphenol oxidase, and catalase were determined. The determination of total phosphorus content adopts the sodium hydroxide melting molybdenum antimony sulfate colorimetric method, the determination of total potassium content adopts the sodium hydroxide melting flame photometer method, the determination of total nitrogen content adopts the elemental analyzer method, the determination of organic carbon content adopts the potassium dichromate volumetric method external heating method, and the determination of pH value adopts the glass electrode method.

### 2.7. Qualitative Analysis of Metabolites and Principal Component Analysis

Metabolite mass spectrometry analysis data for different samples were obtained using Analyst 1.6.3. The peak areas of all mass spectral peaks of the substances were integrated, and the relative content of each component was calculated using the peak area normalization method. Based on the self-built GB-PLANT database, the classification and pathway information of the compounds were identified. Ropls were used for OPLS-DA modeling, VIP values were calculated, and *p* values were calculated using the test. Principal component analysis (PCA) of metabolites was performed using prcomp (R base function). The abscissa of the PCA score chart represents the first principal component, PC1, and the ordinate represents the second principal component, PC2.

### 2.8. Correlation Analysis of Endophytes with Metabolites

Correlation analysis was conducted between metabolites and endophytes. Correlation analysis was conducted between metabolites and soil physics and chemistry using the OmicShare tool (https://www.omicshare.cn). Correlation analysis was conducted between endophyte and soil *Physicochemical index* using RDA (Redundancy analysis).

## 3. Results

### 3.1. OTUs Distribution of Endophytes in P. rotata Roots

The DNA sequences in the root endophytes of *P. rotata* from 4 different habitats (HN, GL, YS, and CD) were analyzed using IlluminaMiSeq sequencing technology (Beijing Baimaike Biotechnology Co., Ltd., Beijing, China). A total bacterias of 1,919,722 pairs of reads were obtained from the 4 habitats after controlling the quality of dual-end reads and splicing (Appendix A). A total bacterias of 1,753,530 clean reads from 12 samples (4 regions, 3 repetitions) were produced with at least 143,549 clean reads per sample and an average of 146,128 clean reads (Appendix A). A total fungus of 1,919,726 pairs of reads was obtained from HN, GL, YS, and CD (Appendix A). After controlling the quality of dual-end reads and splicing, a total fungus of 1,483,529 clean reads was produced with at least 116,768 clean reads per sample and an average of 123,627 clean reads (Appendix A). Differences occur in the number of endophytic OTUs in various habitats. The number of OTUs of endophytic bacteria or fungi in *P. rotata* roots in HN is the highest, with average values of 5115 or 1550, respectively (Appendix A). In addition, compared with those in the other regions, the OTU proportion of endophytic bacteria in the roots of *P. rotata* in YS was relatively low, with an average value of 3049 (Appendix A), while that of endophytic fungi in CD was relatively low, with an average value of 1427 (Appendix A).

A total of 40,644 bacterial OTUs were acquired based on a Venn diagram (Figure 1A). HN and CD had the highest number of common bacterial OTUs with 1339, while YS and CD have the lowest number of common bacterial OTUs with 781. The number of unique bacterial OTUs in HN, GL, YS, and CD were 10,791, 9947, 6745, and 10,006, respectively (Figure 1A). A total of 10,458 fungal OTUs were annotated based on the Venn diagram (Figure 1B). HN and YS had the highest number of common fungal OTUs with 969, while HN and CD had the lowest number of common fungal OTUs with 901. The number of unique fungal OTUs in HN, GL, YS, and CD is 2237, 1962, 2071, and 2058, respectively (Figure 1B). Considerable differences were found in the specific composition of endophytic bacterial populations between YS (6745, Figure 1A) and the 3 other regions (9947–10,791, Figure 1A), while no considerable differences were observed in the specific composition of endophytic fungal populations among the 4 regions with values of 1961–2237 (Figure 1B).

### 3.2. Alpha and Beta Diversity of Endophytes in P. rotata Roots

In alpha diversity comparison, abundance-based coverage estimation (ACE) is an index used to estimate the number of species present in a community. The smaller the value, the lower the richness. The Shannon index and Simpson–Wiener index indicate the diversity of plant communities. The smaller the value, the lower the community diversity. In addition, the coverage rate of endophytic bacteria and fungi was over 99.9%, indicating a reasonable amount of sequencing data for complete coverage (Table 2 and Table 3). Thus, these data represented the endophyte community structure with high confidence, indicating that we can effectively compare endophyte communities in *P. rotata* samples.

The ACE index of bacteria in HN was high, with an average of 5127, while that in YS was low, with an average of 3052 (Table 2). The ACE index of fungi in HN was high at 1561, while that in GL was low at 1430 (Table 3). The Shannon index of bacteria in HN was high at 10.55, while that in YS was low at 7.10 (Table 2). The Shannon index of fungi in HN was high at 9.21, while that in CD was low at 7.43 (Table 3). Therefore, the diversity of bacteria and fungi in HN was higher than in the other regions.

Beta diversity is a comparative analysis of the microbial community composition of different samples. Through nonmetric multidimensional scaling (NMDS) analysis, the classification of multiple samples can be achieved, further demonstrating differences in species diversity among samples. The more similar the compositions of microbial communities in the samples, the closer they are on the coordinate map. With regard to endophytic bacteria, the samples from HN, GL, YS, and CD were separated in the direction of NMSD1 and NMDS2 (Figure 1C). With regard to endophytic fungi, the samples from HN and CD were separated in the direction of NMDS1, while those from GL and HN were separated in the direction of NMSD2 (Figure 1D). The samples from CD were relatively close to the samples from YS in the direction of NMDS2 (Figure 1D).

The unweighted pair group method with arithmetic mean (UPGMA) clustering trees of twelve samples was constructed using four different distance algorithms (Appendix A). The UPGMA clustering trees of endophytic bacteria (Appendix A) and fungi (Appendix A) from *P. rotata* roots in four different habitats were divided into four major branching patterns. These branches group samples with similar community compositions, indicating that the endophytic bacterial and fungal assemblages in *P. rotata* roots were comparable across the four habitat types (Appendix A). At the genus level, the endophytic bacterial abundance of *Pseudomonas*, *Rahnella1*, and *Allorhizobium_Neorhizobium*_*Pararhizobium_Rhizobium* in YS was high, while those in CD were low (Appendix A). With regard to endophytic fungal abundance, unclassified *Helotiales* in GL, *Cladophora* in CD, and *Minimelanolocus* in YS were high (Appendix A).

This result suggested significant differences in the composition of endophytic bacteria in *P. rotata* roots grown in the four distinct environments. In summary, the combined UPGMA clustering and species composition histogram analysis provided a comprehensive assessment of the similarities and differences in endophytic microbial communities associated with *P. rotata* roots across various habitat types. This integrated approach offers valuable insights into the drivers of microbial community structure and assembly within this important plant–microbe system.

### 3.3. Endophytic Community Analysis in P. rotata Roots at the Phylum and Genus Levels

The annotated phyla and genera were statistically analyzed to understand the differences in the microbial composition of *P. rotata* from various habitats (Figure 2, Appendix A). The top 12 endophytic bacteria with relative abundance at the phylum level were as follows: Proteobacteria, Firmicutes, Bacteroidota, Actinobacteriota, Myxococcota, Acidobacteriota, Gemmatimonadota, Desulfobacterota, unclassified bacteria, Fusobacteriota, unassigned, and others (Figure 2A). Among them, 3 endophytic bacteria in all the groups, namely, Proteobacteria, Firmicutes, and Bacteroidota, were the dominant phyla, with the sum of relative abundance accounting for more than 70% of the total relative abundance of all bacterial phyla (Figure 2A). The top 11 endophytic fungi with relative abundance at the phylum level were Ascomycota, Basidiomycota, unclassified fungi, Mortierellomycota, Rozellomycota, Glomeromycota, Chytridiomycota, Olpidiomycota, Mucoromycota, Kickxellomycota, and others (Figure 2B). Among them, 3 endophytic fungi in all the groups, namely, Ascomycota, Basidiomycota, and unclassified fungi, were the dominant phyla, with the sum of relative abundance accounting for more than 70% of the total relative abundance of all fungal phyla (Figure 2B).

At the genus level, the most abundant endophytic bacterial taxa included *Pseudomonas*, *Allorhizobium_Neorhizobium_Pararhizobium*, unidentified*_Lachnospiraceae*, and *Duganella* (Figure 2C). Notably, the relative abundance of the genus *Rahnella1* was the highest in YS compared with that in the other habitats (Figure 2C). Among the abundant fungal genera, *Cadophora* and *Tetracladium* were high in CD (Figure 2D). The unidentified *Helotiales* exhibited high abundance in GL (Figure 2D). *Pseudomonas* and *Rahnella1* exhibited high abundance in YS among endophytic bacteria (Figure 2C). *Minimelanolocus* presented high abundance in YS among endophytic fungi (Figure 2D).

### 3.4. Analysis of the Most Enriched Microorganisms in P. rotata Roots

When the logarithmic linear discriminant analysis (LDA) score for significant differences was set to 4.0 and the number of groups was > 2, LDA effect size could analyze and identify species with significant differences of endophytes from various groups. As shown in Figure 3, the absolute value of the LDA score represents the magnitude of differential species. In CD, the significantly bacterial communities were the p_ Actinobacteriota, c_ Actinobacteria, and f_ Rhodanobacteraceae, f_ Nakamurellaceae, *g_ Nakamurella*, o_Frankiales, *s_ unclassified Actinocorralia*, o_ Thermomonosporae, and o_ Streptosporangiales (Figure 3A and Appendix A). In GL, the significantly enriched bacterial taxa were p_ Firmicutes, o_Burkholderiales, and c_Clostridia, *g_ Duganella*, o_ Lachnospirales, *g_ Haliangium*, o_ Haliangiales, f_ Caulobacteraceae, and o_ Caulobacterales (Figure 3A and Appendix A). In HN, the significantly enriched bacterial taxa were c_ Alphaprotobacter, p_ Myxococcota, and c_ Polyangia, f_ Sphingomonadaceae, o_ Sphingomonadales, and f_ Nitrosomonadaceae (Figure 3A and Appendix A). In YS, the significantly enriched bacterial taxa were f_ Pseudomonas and c_ Gammaproteobacteria, *g_ Allorhombium-Neorhombium-Pararhombium- Rhizobium*, *s_ unclassified_Rahnella1*, and *s_ unclassified_Pseudomonas* (Figure 3A and Appendix A).

With regard to endophytic fungi, the significantly enriched fungal taxa in CD were c_ Leontiomycotes, o_ Helotiales, f_ Phaeosphaeriaceae, *s_Exophiala_tremulae* and f_ Phaeosphaeriaceae (Figure 3B and Appendix A). In GL, the significantly enriched fungal taxa were g_ unclassified Helotiales and f_ unclassified Helotiales (Figure 3B and Appendix A). In HN, the significantly enriched fungal taxa were p_ Basidiomycota, c_ Agaricomycetes, o_ Cantharellales, f_ Clavariaceae, o_ Agaricales, *g_Ceratobasidium*, and f_ Ceratobasideaceae (Figure 3B and Appendix A). In YS, the significantly enriched fungal taxa were g*_ Minimelanolochus*, *s_ Minimelanolochus_obscuru*, and o_ Chaetothyriales (Figure 3B and Appendix A).

In summary, with regard to bacteria, YS, GL, and CD had the most enriched species at the genus level, with 22, 21, and 20, respectively. Meanwhile, HN had the least enriched species at the genus level, with 8. With regard to fungi, CD, HN, and YS had the highest enrichment of 11 species at the genus level, while GL had the lowest enrichment of 3 species. Evidently, more enriched species of endophytic bacteria than fungi were found in *P. rotata* roots at the genus level.

### 3.5. Changes in the Content of Top 30 Metabolites in P. rotata Roots from Four Different Habitats

Through metabolomic analysis, we identified a total of 600 metabolites and selected the top 30 DAMs with the highest content levels for further examination (FC > 1, *p* < 0.05, and VIP > 1). This selection included 5 sugars and alcohols, 4 phenylpropanoids, 7 amino acids, 3 organic acids, 3 alkaloids, 2 terpenoids, 2 polyphenols, 1 nucleotide, 1 ketone, aldehydes, acids, 1 vitamin, and 1 other (Appendix A, Figure 4A). Among them, the vast majority of amino acid compounds were at high levels in regions GL and CD (Figure 4D). In phenylpropane compounds, the level of forsythoside B was high (Figure 4B). Comparing the differential metabolites among four regional groups, the volcano plot showed that the distribution range of differential metabolites between HN and GL was relatively large, while the distribution range of differential metabolites between GL and YS was relatively small (Figure 4E). The volcano plot (Figure 4E) revealed that HN and CD exhibited 15 downregulated and 15 upregulated metabolites, while GL and YS demonstrated 23 downregulated and 7 upregulated metabolites. Additionally, GL and CD displayed 7 downregulated and 23 upregulated metabolites, and HN and GL showed 19 downregulated and 11 upregulated metabolites. HN and YS exhibited 24 downregulated and 6 upregulated metabolites, whereas YS and CD demonstrated 3 downregulated and 27 upregulated metabolites (Figure 4E). A cluster heat map analysis was conducted to visualize the changes in the content of the top 30 metabolites (Figure 4F, Appendix A). The results indicated that the distribution of high-content compounds was specific to the four habitats. Notably, the expression levels of metabolites Forsythoside B and Malic Acid were relatively high in all four regions (Figure 4F). Five compounds with elevated levels were identified in HN, including D-(+)-malic acid, (S)-malic acid, verbascoside, forsythiaside A, and forsythoside I (Figure 4F), while two compounds with elevated levels were identified in GL (Figure 4F). The expression levels of six metabolites were low in all four regions and reached their lowest in YS, including 3-(Carboxymethylamino) propanoic acid, kaempferol-3-O-Β-D-glucosyl (1-2) rhamnoside, L-glutamic acid, shanzhiside methyl ester, 4-hydroxyisoleucine, haplopine, and chlorogenic ccid (Figure 4F). Furthermore, eight compounds were found to have high levels in CD, including L-ornithine (hydrochloride), N-acetylneuraminic acid, dl-Arginine, cyclo (Ile-Leu), 4-methyl-5-thiazoleethanol, stachydrine, isomaltose, and sucrose (Figure 4F).

### 3.6. Determination of Physicochemical Indicators of Rhizosphere Soil in P.rotata Growth Regions

Twelve physicochemical indicators, including amylase, polyphenol oxidase, alkaline protease, alkaline phosphatase, acid protease, acid phosphatase, soil cellulase, sucrase, total nitrogen, total phosphorus, total potassium, and soil organic carbon, were measured and analyzed in the rhizosphere soil of *P. rotata* across different habitats. Among them, soil cellulase and sucrase showed high activity levels in four different habitats, with 41.00 − 6.13 (U/g) and 40.63 − 68.16 (U/g), respectively (Figure 5A). The alkaline phosphatase and acid phosphatase indicated higher activity levels with 20.20 − 26.78 (U/g) and 19.71 − 20.63 (U/g), respectively (Figure 5A). But there is no significant difference in the activity level of acid phosphatase among the four regions (Figure 5A). The activity levels of polyphenol oxidase, acid protease, amylase, and alkaline protease were low in all four regions, with 0.24 − 0.47 (U/g), 0.67 − 4.44 (U/g), 1.17 − 4.24 (U/g), and 1.03 − 6.71 (U/g) (Figure 5A). Four physical and chemical indicators in the soil showed that soil organic carbon had high content in the HN, GL, and CD regions with 45.03 − 78.83 (g/Kg) and the lowest content level in the YS region with 1.47 (g/Kg) (Figure 5B). Total potassium had a high content level in the HN, GL, and CD region with 14.44 − 14.87 (g/Kg), and there is no significant difference in the HN, GL, and CD region (Figure 5B). Total potassium showed a low content level in the YS region with 10.23 (g/Kg). The content levels of total nitrogen and total phosphorylation are relatively low in four habitats, with 0.60 − 1.47 (g/Kg), 0.80 − 1.93 (g/Kg) (Figure 5B).

### 3.7. Relationship Between Endophytic Bacteria or Fungus and Top 30 Metabolites in P. rotata Roots

Different endophytic bacteria exhibit different correlations with different metabolites (Figure 6). We analyzed the relationship between the top 15 bacterial and top 30 metabolite expression levels in the roots of *P. rotata* from four different habitats (Figure 6A). The analysis revealed a total of 109 positive significant correlations and 57 negative correlations (Figure 6A). Notably, the OTUs of *Acidibacter* or *Unclassified_Xanthobacteraceae* exhibited positive correlations with 15 metabolites, such as Stachydrine Hydrochloride, Stachydrine, and Haplopine. The OTUs of *Sphingomonas* showed positive correlations with 10 metabolites, including sucrose, isomaltose, turanose, N-acetylneuraminic acid, L-ornithine (hydrochloride), Dl-arginine, forsythoside B, 4-methyl-5-thiazoleethanol, cyclo(Ile-Leu), and angoroside C (Figure 6A). Furthermore, the OTUs of *Pseudomonas and Allorhizobium_Neorhizobium_ Pararhizobium_Rhizobium* demonstrated negative correlations with 14 metabolites, such as kaempferol-3-O-Β-D-glucosyl (1-2) rhamnoside, 3-(carboxymethylamino) propanoic acid, and L-glutamic acid (Figure 6A). *Variovorax* or *Sphingobium* demonstrated positive correlations with 6 metabolites (shanzhiside methyl ester, 1-phenylpentan-1-one, (S)-malic acid, verbascoside, malic acid, and D-(+)-malic acid), negative correlations with 7 metabolites (N-acetylneuraminic acid, L-ornithine (hydrochloride), Dl-arginine, forsythoside B, 4-methyl-5-thiazoleethanol, cyclo(Ile-Leu) and angoroside C) (Figure 6A).

Similarly, a correlation analysis was performed to investigate the relationship between the top 15 fungal and top 30 metabolite expression levels in *P. rotata* roots (Figure 6B). This analysis identified a total of 59 positive significant correlations and 58 negative correlations (Figure 6B). The OTUs of *Ceratobasidium* was positively correlated with 13 metabolites (such as kaempferol-3-O-Β-D-glucosyl (1-2) rhamnoside, 3-(carboxymethylamino) propanoic acid and L-glutamic acid), while the OTUs of *Tetracladium* or *Cadophora* demonstrated positive correlations with 12 metabolites (such as turanose, isomaltose and 4-methyl-5-thiazoleethanol) (Figure 6B). The OTUs of *Minimelanolocus* demonstrated negative correlations with 11 metabolites (1-phenylpentan-1-one, kaempferol-3-O-Β-D-glucosyl (1-2) rhamnoside, 3-(carboxymethylamino) propanoic acid, and L-glutamic acid) (Figure 6B).

In a word, the bacteria (*Acidibacter*, *Sphingomonas, Variovorax* or *Sphingobium*) and fungus (*Tetracladium, Cadophora or Minimelanolocus*) showed positively or negatively correlated with various secondary metabolites, suggesting a certain relationship between them and the accumulation of metabolites in *P. rotata* roots from four different habitats.

### 3.8. Relationship Between Soil Physical and Chemical Indicators and Top 30 Metabolites in P. rotata

A correlation analysis was conducted to examine the relationship between soil physical and chemical indicators and the levels of the top 30 metabolites in the roots of *P. rotata* from four different habitats (Figure 7). The analysis revealed a total of 59 significant positive correlations and 58 negative correlations (Figure 7). Notably, the content of soil organic carbon was positively correlated with 15 compounds (L-glutamic acid, 3-(carboxymethylamino) propanoic acid, kaempferol-3-O-Β-D-glucosyl(1-2) rhamnoside, 1-phenylpentan-1-one, 4-hydroxyisoleucine, forsythiaside A, shanzhiside methyl ester, haplopine, stachydrine hydrochloride, stachydrine, D-(+)-malic acid, malic acid, verbascoside, chlorogenic acid and turanose), while the content of total potassium in the soil showed positive correlations with 11 compounds (L-glutamic acid, 3-(carboxymethylamino) propanoic acid, kaempferol-3-O-Β-D-glucosyl(1-2) rhamnoside, 1-phenylpentan-1-one, 4-hydroxyisoleucine, forsythiaside A, shanzhiside methyl ester, haplopine, D-(+)-malic acid, malic acid, verbascoside) (Figure 7).

Conversely, the activity of acid protease exhibited negative correlations with 13 compounds (L-glutamic acid, 3-(carboxymethylamino) propanoic acid, kaempferol-3-O-Β-D-glucosyl(1-2) rhamnoside, 1-phenylpentan-1-one, 4-hydroxyisoleucine, forsythiaside A, shanzhiside methyl ester, haplopine, stachydrine hydrochloride, stachydrine, D-(+)-malic acid, malic acid, verbascoside), and alkaline phosphatase demonstrated negative correlations with 9 compounds (turanose, isomaltose, angoroside C, cyclo(Ile-Leu), 4-methyl-5-thiazoleethanol, forsythoside B, Dl-arginine, L-ornithine (hydrochloride), N-acetylneuraminic acid). The activity of amylase exhibited 13 positive correlations with 11 compounds (chlorogenic acid, verbascoside, malic acid, D-(+)-malic acid) and negative correlations with 2 compounds (angoroside C, 2′-deoxyadenosine). Notably, the expression of soil organic carbon and total potassium were positively correlated with 6 compounds (L-glutamic acid, 3-(carboxymethylamino) propanoic acid, kaempferol-3-O-Β-D-glucosyl (1-2) rhamnoside, 1-phenylpentan-1-one, 4-hydroxyisoleucine, forsythiaside A), while the activity of acid protease were negatively correlated with them (Figure 7).

### 3.9. Relationship Between Endophytic Bacteria or Fungus and Soil Physical and Chemical Indicators in P. rotata

A correlation analysis was conducted to examine the relationship between the top 15 bacteria and the soil’s physical and chemical indicators across four distinct habitats in the roots of *P. rotata* (Figure 8A). The analysis revealed a total of 33 significant positive correlations and 33 significant negative correlations (Figure 8A). The OTUs of *Acidibacter* were significantly positively correlated with 5 soil physicochemical indicators (total phosphorus, amylase, sucrase, total potassium or soil organic carbon) and significantly negatively correlated with acid protease (Figure 8A). *Pseudomonas* was significantly positively correlated with acid protease and total nitrogen, along with a significant negative correlation with total potassium and Soil organic carbon (Figure 8A).

Similarly, a correlation analysis was performed to investigate the relationship between the top 15 fungi and soil physical and chemical indicators in the roots of *P. rotata* across the same four habitats (Figure 8B). The results indicated a total of 25 significant positive correlations and 17 significant negative correlations (Figure 8B). The OUTs of *Minimelanolocus* showed positive correlations with acid protease, total nitrogen, acid phosphatase, or soil cellulase and negative correlations with total potassium or soil organic carbon (Figure 8B). *Tetracladium* or *Cadophora* showed positive correlations with total phosphorus and negative correlations with alkaline phosphatase (Figure 8B).

The distance-based redundancy analysis (db-RDA) based on Bray-Curtis distance revealed that community and the levels of soil physical and chemical indicators in the roots of *P. rotata*, acid protease, total nitrogen, acid phosphatase, alkaline phosphatase, and alkaline protease (R^2^ = 0.9869, *p* < 0.001) were crucial environmental driving factors in YS region (Figure 8C). Total phosphorus, sucrase, and soil cellulase were crucial factors in the CD region, and total potassium and amylase in the HN region (Figure 8C). These factors influenced the distribution of endophytic bacteria communities, resulting in RDA1 45.65% and RDA2 28.71% of the overall variability in the composition of endophytic bacteria communities (Figure 8C).

The db-RDA based on Bray-Curtis distance revealed that fungus community and the content levels of soil physical and chemical indicators in the roots of *P. rotata* (R^2^ = 0.8957, *p* < 0.001). The fungal community OTUs in the HN and GL regions are similar. Total potassium and amylase were crucial environmental driving factors in the HN and GL regions, and total phosphorus, polyphenol oxidase, and sucrase in the CD region (Figure 8D). Alkaline phosphatase, acid phosphatase, acid protease, alkaline protease, and total nitrogen were crucial environmental driving factors in the YS region (Figure 8D). These factors influenced the distribution of endophytic fungus communities, resulting in RDA 1 47.04% and RDA2 41.05% of the overall variability in the composition of endophytic fungus communities (Figure 8D).

## 4. Discussion

### 4.1. Endophytic Bacteria and Fungi of P. rotata Roots

Plant-associated microbiota play a pivotal role in regulating various biological processes, influencing various characteristics of plant growth, development, and responses to environmental stressors [39,40]. Understanding the community structure and interspecific interactions between root-associated bacteria/fungi and their host plants is crucial for elucidating these symbiotic relationships [41]. The endophytic Ascomycota orders *Helotiales* and *Chaetothyriales* displayed distinct patterns of host associations, with many taxa shared as primary symbionts across multiple plant species. Endophytic bacteria and fungi are ubiquitous within a diverse range of plant species, influencing their growth, development, and metabolic processes [42,43].

In the current study, we employed high-throughput sequencing to analyze the community composition of endophytic microorganisms that inhabit *P. rotata* roots from four different habitats. In this study, the dominant bacterial genera were *Pseudomonas*, *Allorhizobium*, unidentified *Lachnospiraceae*, and *Duganella*, while the dominant fungal taxa were unidentified fungi and unidentified *Helotiales*. Furthermore, endophytic bacteria and fungi represent highly promising components for the development of effective biofertilizers, biostimulants, and biocontrol agents. Therefore, a concerted effort to isolate and characterize endophytic bacteria from diverse plant sources is warranted, along with investigations into the conditions that govern their plant growth-promoting capabilities.

### 4.2. Relationship Between Endophytic Bacteria or Fungus and Top 30 Metabolites in P. rotata

Microorganisms can enhance secondary metabolism by interacting directly or indirectly with their hosts, although some secondary metabolic processes are exclusive to the eukaryotic domain [44]. A total of 51 fungal endophytes were isolated from four PhG-producing plants: *Echinacea purpurea*, *Rehmannia glutinosa*, *Ligustrum lucidum*, and *Cistanche deserticola*. The endophyte Simplicillium sinense EFF1, derived from *Echinacea purpurea*, demonstrated the capacity to de-rhamnose isoacteoside, resulting in the production of calceorioside B, a multifunctional PhG derivative [45]. Li et al. investigated the relationship between endophytic diversity and the metabolites produced across different tissues of *Panax quinquefolius* [46]. At the phylum level, Cyanobacteria emerged as the dominant endophytic bacteria in the roots, fibrils, stems, and leaves of P. quinquefolius [46]. Conexibacter was notably enriched in the roots and fibrils, exhibiting a significant positive correlation with saponin differential metabolites. Conversely, *Cyberlindnera* was significantly enriched in the stems and leaves, displaying a notable negative correlation with differential metabolites (*p* < 0.05) [46]. Yang et al. selected *Paeonia lactiflora* (with the medicinal cultivar known as ’Hangbaishao’ or HS and the ornamental cultivar as ’Zifengyu’ or ZFY) to analyze the microbiome and metabolome [47]. The key bacterium, Ruminococcaceae bacterium GD7, was found to facilitate the accumulation of phenolic acids and flavonoids in ZFY [47]. With water-soluble humic materials treatment, *Sphingobium*. TBBS4 was isolated from soybean and significantly improved soybean nodulation and growth by increasing della gene expression and reducing ethylene release [48]. An aquatic *Tetracladium*, isolated as a root endophyte from riparian plants in Nainital, Kumaun Himalaya, India, was characterized by laterally applanate conidia appearing lobate, with typically four rounded apices and lacking filiform, acicular, or subulate elements [49]. A study identified the volatile metabolites associated with three distinct diseases—stem rot, blue mold, and green mold—in citrus fruits post-harvest that indicated eight volatile compounds were identified as biomarkers for citrus stem rot, while one compound served as a biomarker for citrus green mold, effectively distinguishing infected citrus fruits [50].

In our study, correlation analysis was performed to investigate the relationship between the top 15 bacteria or fungus and top 30 metabolites expression levels from four different habitats in *P. rotata* roots (Figure 6). The bacteria (*Sphingomonas* and *Acidibacter* or *Unclassified_Xanthobacteraceae*) exhibited positive correlations with most of the compounds. Additionally, the expression of fungus (*Ceratobasidium*, *Tetracladium,* or *Cadophora)* exhibited positive correlations with most of the compounds. Therefore, we infer that they play an important role in the enrichment process of metabolites in *P. rotata* roots.

### 4.3. Relationship Between Endophytic Bacteria or Fungus and Soil Physical and Chemical Indicators in P. rotata

Plant interactions with endophytic bacteria yield mutual benefits and contribute to environmental sustainability [51]. The dominant genera exhibiting significant distribution differences among these plant tissue samples included *Burkholderia-Caballeronia-Paraburkholderia*, *Sphingomonas*, *Acidibacter*, *Bradyrhizobium*, *Bryobacter, Methylocella, Nocardioides, Acidothermus,* and *Allorhizobium-Neorhizobium-Pararhizobium-Rhizobium* [52]. A study characterized three *Alternaria* endophytic fungi isolated from their host, Nettle (*Urtica dioica* L.). The antibacterial activity was evaluated against reference and isolated strains, including Methicillin-Resistant Staphylococcus aureus. A wide range of antimicrobial activities, similar to those measured in nettle leaves, was detected, particularly for *Alternaria sorghi* [53]. Both Byssochlamys spectabilis and Alternaria sp. extracts exhibited antibacterial activity against the *S. aureus* strain [53]. The findings of this study suggest that endophytic fungi associated with medicinal plants from Sudan may serve as a promising source of new therapeutic compounds.

In our study, correlation analysis was performed to investigate the relationship between the top 15 bacteria or fungus and soil physical and chemical indicators’ expression levels from four different habitats in *P. rotata* roots (Figure 8A). The expression of bacteria (*pseudomonase*) was significantly positively correlated with 5 soil physicochemical indicators and significantly negatively correlated with 2 soil physicochemical indicators. Additionally, the expression of fungus (*Minimelanolocus*) exhibited positive correlations with 5 soil physicochemical indicators and significantly negatively correlated with 2 soil physicochemical indicators. Therefore, we infer that both endophytic bacteria and soil physicochemical factors can affect the accumulation of metabolites in *P. rotata* roots.

## 5. Conclusions

We conducted endophyte and metabolite identification and characterized the correlation in *P. rotata* roots from four different habitats at an altitude of 3500–4300 m. The top 30 DAMs with high content levels were screened. Total potassium only exhibited a low content level, specifically in the YS region. We analyzed the relationship between the top 15 bacterial and top 30 metabolite expression levels in the roots of *P. rotata* from four different habitats. The analysis revealed a total of 109 positive significant correlations and 57 negative correlations. Notably, the OTUs of *Acidibacter* or *Unclassified_Xanthobacteraceae* exhibited positive correlations with 15 metabolites, such as Stachydrine Hydrochloride, Stachydrine, and Haplopine. The OTUs of *Sphingomonas* showed positive correlations with 10 metabolites, including sucrose, isomaltose, turanose, N-acetylneuraminic acid, L-ornithine (hydrochloride), Dl-arginine, forsythoside B, 4-methyl-5-thiazoleethanol, cyclo(Ile-Leu) and angoroside C. Similarly, a correlation analysis was performed to investigate the relationship between the top 15 fungal and top 30 metabolite expression levels in *P. rotata* roots. This analysis identified a total of 59 positive significant correlations and 58 negative correlations. The OTUs of *Ceratobasidium* was positively correlated with 13 metabolites (such as kaempferol-3-O-Β-D-glucosyl (1-2) rhamnoside, 3-(carboxymethylamino) propanoic acid and L-glutamic acid), while the OTUs of *Tetracladium* or *Cadophora* demonstrated positive correlations with 12 metabolites (such as turanose, isomaltose and 4-methyl-5-thiazoleethanol). In the YS region, five soil physicochemical indicators are related to endophytic bacteria and fungi associated with *P. rotata* roots. In summary, this research provides a global perspective on the connection between endophytes and active medicinal compounds in *P. rotata* roots, contributing to our understanding of the complex molecular relationships that underpin plant microbiomic and metabolomic adaptations to varying environments.

## Figures and Tables

**Figure 1 microorganisms-13-00503-f001:**
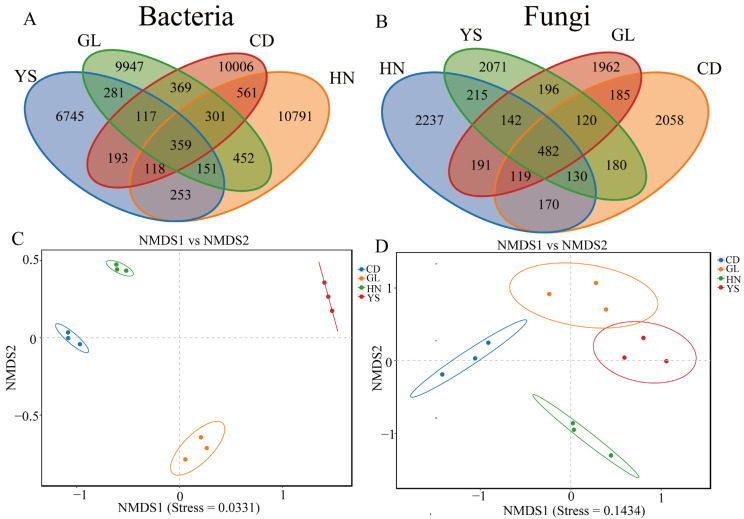
Venn and NMDS diagram of OTU distribution of endophyte in *P. rotata* roots from four habitats. (**A**) Venn diagram of endophytic bacteria. (**B**) Venn diagram of endophytic fungi. (**C**) NMDS diagram of endophytic bacteria. (**D**) NMDS diagram of endophytic fungi.

**Figure 2 microorganisms-13-00503-f002:**
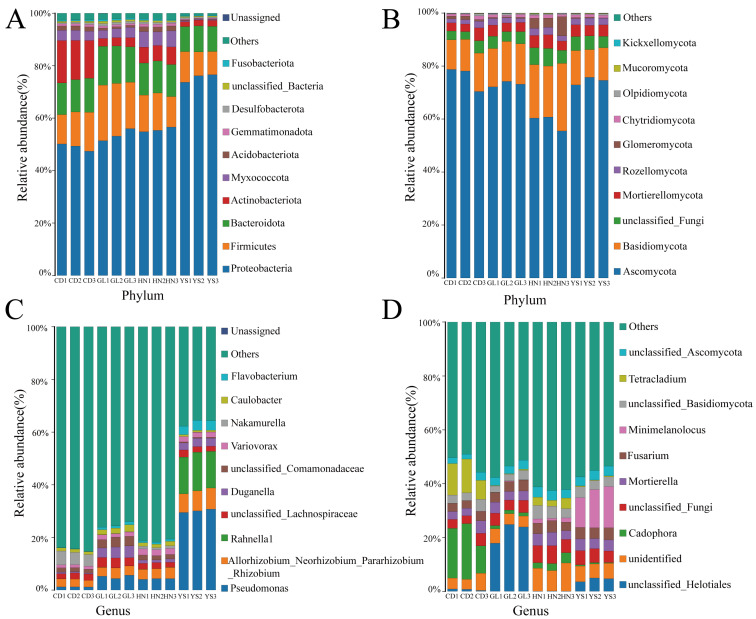
The community and relative abundance of endophytes in *P. rotata* roots from HN, GL, YS, and CD habitats. (**A**) Endophytic bacteria at the phylum level. (**B**) Endophytic fungi at the phylum level. (**C**) Endophytic bacteria at the genus level. (**D**) Endophytic fungi at the genus level. The color legend followed the same order as the colors in the bars, from bottom to top.

**Figure 3 microorganisms-13-00503-f003:**
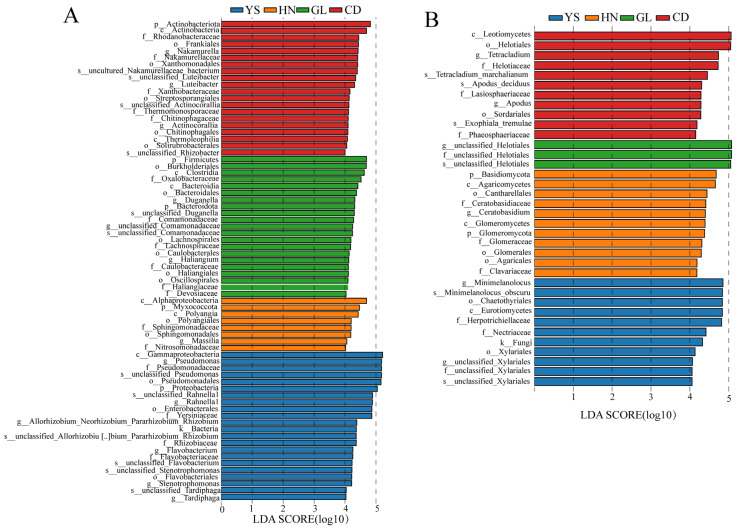
LDA analysis of endophytic communities in P. rotata roots from four habitats. (**A**) LDA analysis of endophytic bacteria at the genera level. (**B**) LDA analysis of endophytic fungi at genera level. The first letter before each bacterial and fungal name represents: p_ represents phylum, o_ represents order, f_ represents family, g_ represents genus, s_ represents species.

**Figure 4 microorganisms-13-00503-f004:**
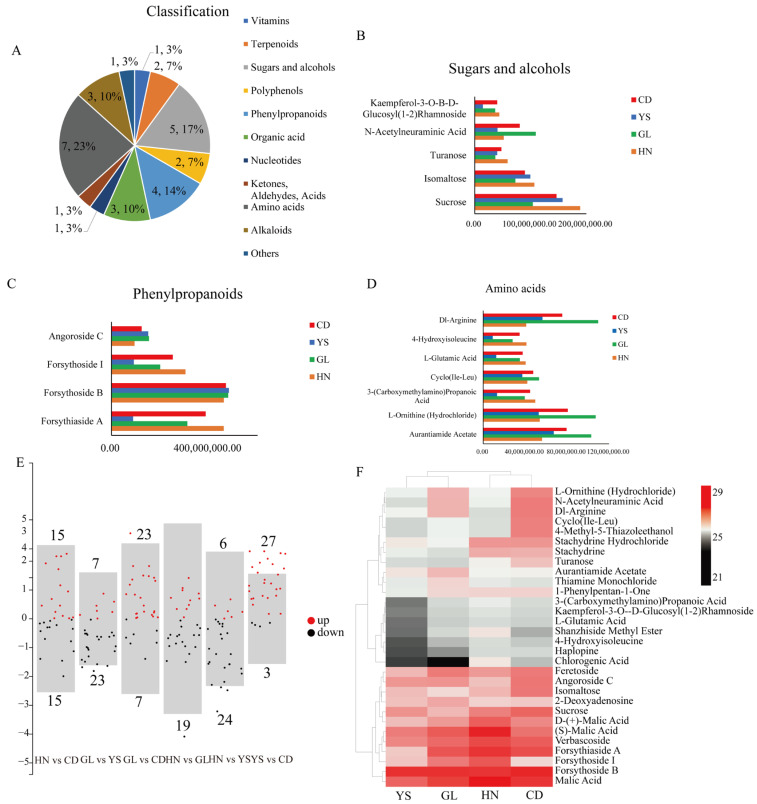
Cluster heatmaps of top 30 metabolomic analyses in *P. rotata* roots from four habitats. (**A**) Classification and proportion of 30 DAMs. (**B**) 5 sugars and alcohol compounds. (**C**) 4 phenylpropanoids compounds. (**D**) 7 amino acid compounds. (**E**) Volcano Map of the distribution of 30 DAMs. The number upper bar represented upregulated-metabolite number, the number below bar represented downregulated-metabolite number. (**F**) Heatmap of 30 DAMs. DAMs: Differentially Accumulated Metabolites.

**Figure 5 microorganisms-13-00503-f005:**
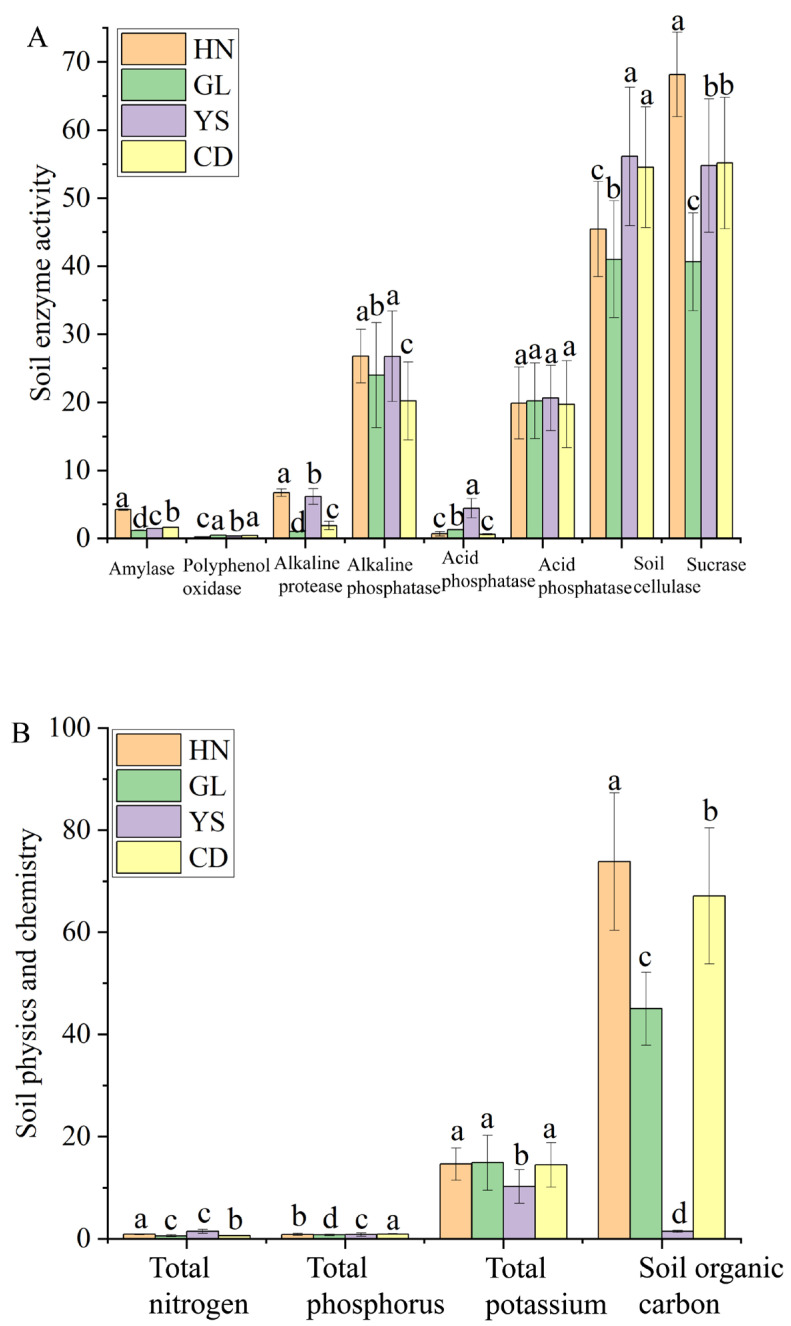
Soil physical and chemical indicators in *P. rotata* roots from four habitats. (**A**) Soil enzyme activity. (**B**) Soil physics and chemistry. Lowercase letters represent differences at the 0.05 level.

**Figure 6 microorganisms-13-00503-f006:**
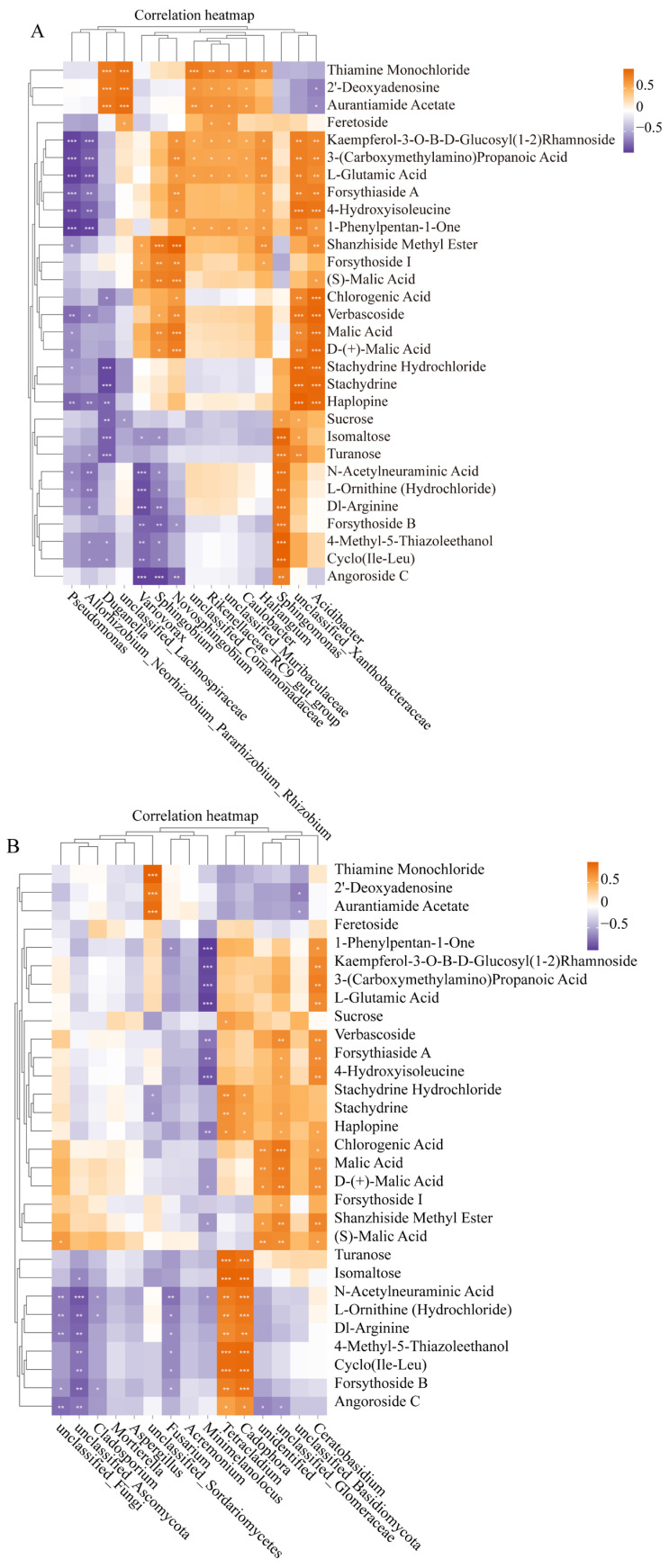
Correlation analysis between top 15 endophytic bacteria or fungus and top 30 metabolites in *P. rotata* roots from four habitats. (**A**) Correlation analysis between 15 bacteria and top 30 metabolites. (**B**) Correlation analysis between 15 fungus and top 30 metabolites. Orange text represented positive relationship, purple text represented negative relationship. * *p* < 0.05, ** *p* < 0.01, *** *p* < 0.001.

**Figure 7 microorganisms-13-00503-f007:**
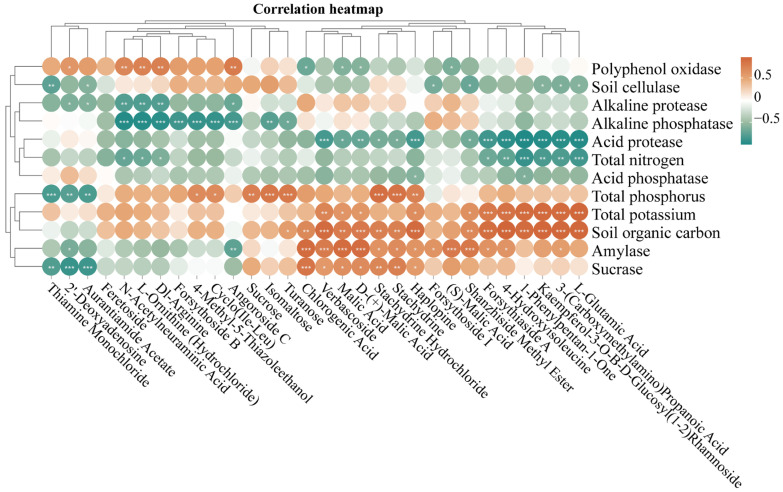
Correlation analysis between soil physical and chemical indicators and top 30 metabolites in *P. rotata* roots from four habitats. Orange text represented positive relationship, green text represented negative relationship. * *p* < 0.05, ** *p* < 0.01, *** *p* < 0.001.

**Figure 8 microorganisms-13-00503-f008:**
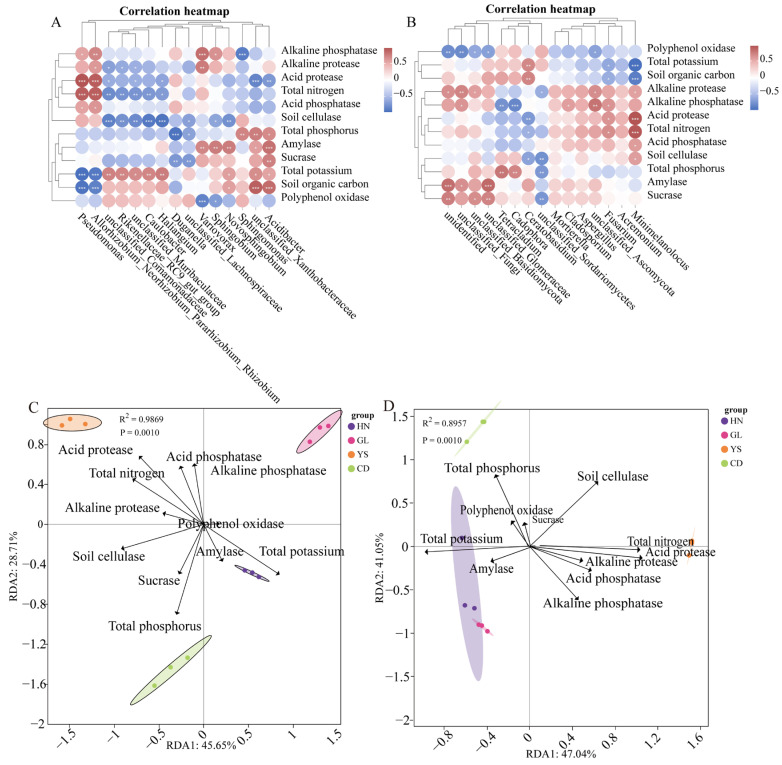
Correlation analysis between top 15 bacteria or fungus and soil physical and chemical indicators in *P. rotata* roots from four habitats. (**A**) Correlation analysis between 15 bacteria and soil physical and chemical indicators. (**B**) Correlation analysis between 15 fungus and soil physical and chemical indicators. Pink text represented positive relationships; blue text represented negative relationship. * *p* < 0.05, ** *p* < 0.01, *** *p* < 0.001. (**C**) RDA correlation between 15 endophytic bacteria and soil physical and chemical indicators from four different habitats. (**D**) RDA correlation between 15 endophytic fungus and soil physical and chemical indicators from four different habitats. The black rays in the figure represent different soil environmental factors, and the length of the rays represents the degree of influence (explanatory power) of the environmental factor on endophytic bacteria or fungus. The angle between environmental factor rays represents the positive or negative correlation (acute angle: positive correlation; obtuse angle: negative correlation; right angle: no correlation).

**Table 1 microorganisms-13-00503-t001:** *P. rotata* samples from 4 habitats.

Sample	Location	Altitude	East Longitude	North Latitude
HN	Henan County, Qinghai Province	3540 m	101°56′48″	34°76′12″
GL	Guoluo County, Qinghai Province	3750 m	100°14′38″	34°29′10″
YS	Yushu County, Qinghai Province	3880 m	97°1′23″	32°51′4″
CD	Chengduo County, Qinghai Province	4270 m	97°27′16″	33°18′2″

**Table 2 microorganisms-13-00503-t002:** Alpha diversity analysis of endophytic bacteria in *P. rotata* roots from four habitats.

Sample	ACE (Mean ± SD)	Chao1 (Mean ± SD)	Simpson (Mean ± SD)	Shannon (Mean ± SD)	Coverage
HN	5128.08 ± 38.86	5124.37 ± 39.04	0.99 ± 0.0001	10.55 ± 0.09	0.9998
GL	4445.89 ± 162.55	4443.25 ± 163.61	0.99 ± 0.0004	10.22 ± 0.15	0.9999
YS	3052.60 ± 131.73	3049.57 ± 130.88	0.95 ± 0.0035	7.2 ± 0.21	0.9999
CD	4657.31 ± 33.70	4654.49 ± 32.77	0.99 ± 0.0007	10.05 ± 0.16	0.9998

ACE: Abundance-based Coverage Estimator. Chao1: Chao1 Index. Simpson: Simpson’s Diversity Index. Shannon: Shannon-Wiener Index or Shannon Entropy.

**Table 3 microorganisms-13-00503-t003:** Alpha diversity analysis of endophytic fungi in *P. rotata* roots from four habitats.

Sample	ACE (Mean ± SD)	Chao1 (Mean ± SD)	Simpson (Mean ± SD)	Shannon (Mean ± SD)	Coverage
HN	5161.53 ± 27.37	1560.39 ± 26.02	0.99 ± 0.002	8.88 ± 0.35	0.9998
GL	1418.20 ± 34.05	1419.54 ± 35.74	0.94 ± 0.016	7.56 ± 0.39	0.9998
YS	1473.47 ± 123.96	1471.63 ± 42.74	0.97 ± 0.006	8.23 ± 0.25	0.9998
CD	1437.46 ± 43.21	1436.53 ± 42,74	0.96 ± 0.021	7.44 ± 0.85	0.9998

ACE: Abundance-based Coverage Estimator. Chao1: Chao1 Index. Simpson: Simpson’s Diversity Index. Shannon: Shannon-Wiener Index or Shannon Entropy.

## Data Availability

The endophytic bacteria raw RNA-seq datasets can be found in the NCBI SRA under the project number: PRJNA1134102. https://www.ncbi.nlm.nih.gov/bioproject/PRJNA1134102/ (Released on 23 October 2024). The endophytic fungi raw RNA-seq datasets can be found in the NCBI SRA under the project number: PRJNA1134636. https://www.ncbi.nlm.nih.gov/bioproject/PRJNA1134636/ (Released on 23 October 2024).

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
