# Peer review of "Diversity and Correlation Analysis of Endophytes and Top Metabolites in *Phlomoides rotata* Roots from High-Altitude Habitats"

_microorganisms, 2025, doi:10.3390/microorganisms13030503_

Round 1

Reviewer 1 Report

Comments and Suggestions for Authors

Dear authors,

First of all, congratulations on your work titled Diversity and Correlation Analysis of Endophytes and top Metabolites in Phlomoides rotata Roots from High-altitude Habitats.  The manuscript effectively fulfills the methodology and results required to support the objective of determining the composition and diversity of endophytes in the four mentioned habitats, differentiated by their altitude above sea level. I am sharing some comments that I believe should be reviewed before accepting its publication.

Introduction:

-          Line 87: The first time abbreviations or acronyms are used, their meaning should be explained. It is necessary to add the meaning and correctly write pH.

-          Line 195: In the sentence, the missing punctuation marks should be added.

Material and Methods:

-          Lines 107 & 110: If it refers to the same four habitats, please improve the wording.

-          Lines 130 and beyond: please improve the wording. Methodologies are often written as a past activity. Just line 2.1 and 2.3 sections of the manuscript.

-          Line 175: Considering the type of analysis being described, it is suggested to explain what the concept of “conducted selected score” refers to. The authors are encouraged to improve the writing of Section 2.6 in its entirety, as the activities carried out to obtain the soil samples and the subsequent enzymatic analysis are not clearly described.

Results:

-          Figure 2: The authors are advised to ensure that, in the figures, the color legend follows the same order as the colors in the bars. It should indicate the composition at the genus level from highest to lowest.

-          Line 314: What the author mean when the title of the section 3.4 indicates analysis of “superiority” endophytic bacteria? It is suggested to use a synonym to describe what are you showing in the main text. Maybe are you trying to describe microbial enrichment?

-          Line 383: The quality of Figure 4 does not allow the provided information to be read clearly. The authors are encouraged to improve it to better appreciate the presented results.

In the conclusions, beyond describing the main result for each of the four habitats, it would be important to clarify how the distribution of metabolites varied (increased or decreased) in association with the main representatives of bacteria or fungi in each environment.

Comments on the Quality of English Language

Dear authors,

Although the manuscript is written in English that allows for proper understanding, there are some paragraphs where the writing style and the selection of certain technical terms should be improved. This is because it becomes challenging to understand what the authors aim to convey. Furthermore, the terms used suggest meanings different from what is intended. For this reason, the authors are encouraged to seek an English language review before submitting their revised work.

Author Response

Comments and Suggestions for Authors

Dear authors,

First of all, congratulations on your work titled Diversity and Correlation Analysis of Endophytes and top Metabolites in Phlomoides rotata Roots from High-altitude Habitats.  The manuscript effectively fulfills the methodology and results required to support the objective of determining the composition and diversity of endophytes in the four mentioned habitats, differentiated by their altitude above sea level. I am sharing some comments that I believe should be reviewed before accepting its publication.

Introduction:

Q1-Line 87: The first time abbreviations or acronyms are used, their meaning should be explained. It is necessary to add the meaning and correctly write pH.

Reply 1: We added the full name on line 88-89. We have also checked and modified the full names and abbreviations in the whole paper.

Q2-Line 195: In the sentence, the missing punctuation marks should be added.

Reply 2: We added punctuation marks on line 196.

Material and Methods:

Q3-Lines 107 & 110: If it refers to the same four habitats, please improve the wording.

Reply 3: Endophytes in P. rotata roots from four different habitats were analyzed. Four different habitats were HN, YS, CD, and GL regions.

Q4-Lines 130 and beyond: please improve the wording. Methodologies are often written as a past activity. Just line 2.1 and 2.3 sections of the manuscript.

Reply 4: We have made modifications to this content in“2 Materials and Methods”, line 132-143, 179-191.

Q5-Line 175: Considering the type of analysis being described, it is suggested to explain what the concept of “conducted selected score” refers to. The authors are encouraged to improve the writing of Section 2.6 in its entirety, as the activities carried out to obtain the soil samples and the subsequent enzymatic analysis are not clearly described.

Reply 5: We deleted the “conducted selected score” in section 2.6, line 176. We added the contents in line 179-192, “2.6 the Soil Sample Collection and Physicochemical Property Measurements”,

Q6-Figure 2: The authors are advised to ensure that, in the figures, the color legend follows the same order as the colors in the bars. It should indicate the composition at the genus level from highest to lowest.

Reply 6: The caption has been added to Figure 2. The color legend followed the same order as the colors in the bars from bottom to top.

.

Q7-Line 314: What the author mean when the title of the section 3.4 indicates analysis of “superiority” endophytic bacteria? It is suggested to use a synonym to describe what are you showing in the main text. Maybe are you trying to describe microbial enrichment?

Reply 7: We have modified “3.4 Analysis of the most enriched microorganisms in P. rotata roots” on Line 324.

Q8-Line 383: The quality of Figure 4 does not allow the provided information to be read clearly. The authors are encouraged to improve it to better appreciate the presented results.

In the conclusions, beyond describing the main result for each of the four habitats, it would be important to clarify how the distribution of metabolites varied (increased or decreased) in association with the main representatives of bacteria or fungi in each environment.

Reply 8: We have modified and enlarged Figure 4.

In the conclusion, we added the correlation between metabolites and the main representatives of bacteria or fungi in line 619-634.

Reviewer 2 Report

Comments and Suggestions for Authors

This study investigates the diversity and relationship between endophytes and metabolites in the roots of Phlomoides rotata from high-altitude habitats on the Qinghai-Tibet Plateau. Using 16S and ITS2 sequencing and metabolomics, the authors analysed the endophytic communities and identified 30 metabolites, including amino acids, sugars and phenylpropanoids. Significant positive and negative correlations were observed between certain endophytic genera (e.g. Acidibacter, Sphingomonas, Tetracladium, Cadophora) and metabolites. In addition, the correlations between soil physico-chemical properties and metabolite levels emphasised the influence of environmental factors. This research provides insights into plant-microbiome interactions and their impact on the synthesis of medicinal metabolites. The manuscript is an interesting and valuable contribution to the understanding of the relationship between endophytes, metabolites and environmental factors in a medicinal plant. However, minor corrections and clarifications are needed to improve readability and scientific rigour.

Lines 27, 43, 76, 95, 177 – Please correct the typos.

Line 130 – The wording of this sentence is unclear; please rephrase it for clarity.

Lines 134, 178 – Avoid writing the methods section like a manual or handbook. Describe the procedures in the past tense.

Line 147 – Replace "is used" with "has been used" and ensure that the past tense is used consistently throughout the methods section.

Table 1 is missing. Make sure it is included before continuing with Table 2.

Replace Tables 2 and 3 with a figure showing bar charts of averages and standard deviations (SD) for all three replicates.

Figure 3 – The labels are too small to read. Consider reducing the data shown and focus on the most common taxa. Include the full data set in the supplementary materials.

Figures 4 and 6 – Increase the font size for more consistency.

Figure 5 – Add replicates and error bars. If no replicates are available, justify this and adjust your data interpretation.

Author Response

Comments and Suggestions for Authors

This study investigates the diversity and relationship between endophytes and metabolites in the roots of Phlomoides rotata from high-altitude habitats on the Qinghai-Tibet Plateau. Using 16S and ITS2 sequencing and metabolomics, the authors analysed the endophytic communities and identified 30 metabolites, including amino acids, sugars and phenylpropanoids. Significant positive and negative correlations were observed between certain endophytic genera (e.g. AcidibacterSphingomonasTetracladiumCadophora) and metabolites. In addition, the correlations between soil physico-chemical properties and metabolite levels emphasised the influence of environmental factors. This research provides insights into plant-microbiome interactions and their impact on the synthesis of medicinal metabolites. The manuscript is an interesting and valuable contribution to the understanding of the relationship between endophytes, metabolites and environmental factors in a medicinal plant. However, minor corrections and clarifications are needed to improve readability and scientific rigour.

Q1 Lines 27, 43, 76, 95, 177 – Please correct the typos.

Reply 1: We have modified this section on lines 30, 43, 77, 97, 177.

Q2 Line 130 – The wording of this sentence is unclear; please rephrase it for clarity.

Reply 2: We have modified this sentence on line 131-132.

Q3 Lines 134, 178 – Avoid writing the methods section like a manual or handbook. Describe the procedures in the past tense.

Reply 3: We have modified this sentence on line 132,179-191.

Q4 Line 147 – Replace "is used" with "has been used" and ensure that the past tense is used consistently throughout the methods section.

Reply 4: We have modified this sentence on Line 152.

Q5 Table 1 is missing. Make sure it is included before continuing with Table 2.

Replace Tables 2 and 3 with a figure showing bar charts of averages and standard deviations (SD) for all three replicates.

Reply 5: Table 1. on Line 128. Tables 2 and 3 have been modified.

Q6 Figure 3 – The labels are too small to read. Consider reducing the data shown and focus on the most common taxa. Include the full data set in the supplementary materials.

Reply 6: The original Figure 3A, 3B has been enlarged. The original Figure 3C, 3D has been changed into Figure S2.

Q7 Figures 4 and 6 – Increase the font size for more consistency.

Reply 7: We have revised Figures 4 and 6.

Q8 Figure 5 – Add replicates and error bars. If no replicates are available, justify this and adjust your data interpretation.

Reply 8: We made three repetitions for each region, redrawn Figure 5, and added error bars.

Reviewer 3 Report

Comments and Suggestions for Authors

Dear Authors,

The publication is well prepared. The study covers a broader approach to biodiversity in P. rotata roots and possible interactions with the environment. It was interesting to read and review.

A few notes to the authors:

  1. In line 577 and line 588, italic font should be removed from or and and.
  2. In the conclusions, I missed the results on the impact of growth height on endophytes diversity in P. rotata roots. Please add it.
  3. Reviewer's reflection. I regret that the authors did not perform a shotgun metagenomic analysis, in which case a more detailed analysis of endophyte diversity and influence on metabolomics would be possible (at the species level). Also, the diversity of endophytic archaea would be shown, which also has a significant impact on the synthesis of metabolites in the plant or in the plant rhizosphere.

Author Response

Comments and Suggestions for Authors

Dear Authors,

The publication is well prepared. The study covers a broader approach to biodiversity in P. rotata roots and possible interactions with the environment. It was interesting to read and review.

A few notes to the authors:

  1. In line 577 and line 588, italic font should be removed from or and and.

Reply 1: We have already made modifications on line 578, line 589.

  1. In the conclusions, I missed the results on the impact of growth height on endophytes diversity in P. rotata roots. Please add it.

Reply 2: We collected wild P. rotata roots and did not measure the plant height at that time, so we did not analyze this part of the content in the article.

  1. Reviewer's reflection. I regret that the authors did not perform a shotgun metagenomic analysis, in which caserhytic archaea would be shown, which also has a significant impact on the synthesis of metabolites in the plant or in the plant rhizosphere.

Reply 3: The method of metagenomic determination has been supplemented in the materials and methods section, line 132-143. We only conducted analysis at the level of phylum and genus, not archaea.

Reviewer 4 Report

Comments and Suggestions for Authors

Dear authors,

The manuscript is quite detailed; however, it contains significant taxonomic errors. I reviewed lines 319-328 and found numerous mistakes in taxonomy. Please review the following points and make the necessary corrections. I encourage you to apply the same corrections throughout the main text.

Lines 319 – 320

The Actinobacteriota (previously Actinobacteria, (Whitman et al., 2018) is a bacterial phylum

Actinobacteria is a phylum of Gram-positive bacteria

Rhodanobacteraceae is a family of bacteria of the order Xanthomonadales

Line 321

The Nakamurellaceae originally comprised the genera, Nakamurella

The Frankiales are an order of bacteria

Actinocorralia is a genus of the Family Thermomonosporaceae? (please check, I m note sure)

Thermomonosporae the correct name is Thermomonospora

The Streptosporangiales is an order of bacteria

Lines 323-324

Bacillota (synonym Firmicutes) is a phylum of bacteria NOT a genus

Burkholderiales is an order of Betaproteobacteria in the phylum Pseudomonadota NOT a genus

The Clostridia are a highly class of Bacillota (synonym Firmicutes)

Duganella is a genus in the Oxalobacteraceae family

Lachnospirales I think is an order please check

Haliangium is a genus of bacteria from the family of Kofleriaceae ? please check

Haliangiales is the order of the genus Haliangium

Caulobacteraceae is a family of a Phylum Proteobacteria

Caulobacterales is an order

Lines 325-326

Alphaprotobacter the correct is Alphaproteobacteria or α-proteobacteria is a class of bacteria in the phylum Pseudomonadota (formerly "Proteobacteria"), please check

The Myxococcota is a phylum of bacteria, please check

Polyangia, I think the correct name is Polyangium a genus in the family Microscaphidiidae

The Sphingomonadales is an order of the Alphaproteobacteria includes the family Sphingomonadaceae, please check

Nitrosomonadaceae, the correct name is Nitrosomonadaceae. Nitrosomonadaceae is a family of betaproteobacteria

Line 328

Gammaproteobacteria is a class of bacteria

Allorhombium etc are genus of the family Rhizobiaceae? Pleace check

Author Response

Comments and Suggestions for Authors

Dear authors,

The manuscript is quite detailed; however, it contains significant taxonomic errors. I reviewed lines 319-328 and found numerous mistakes in taxonomy. Please review the following points and make the necessary corrections. I encourage you to apply the same corrections throughout the main text.

Q1: Lines 319 – 320

The Actinobacteriota (previously Actinobacteria, (Whitman et al., 2018) is a bacterial phylum

Actinobacteria is a phylum of Gram-positive bacteria

Rhodanobacteraceae is a family of bacteria of the order Xanthomonadales

Q2: Line 321

The Nakamurellaceae originally comprised the genera, Nakamurella

The Frankiales are an order of bacteria

Actinocorralia is a genus of the Family Thermomonosporaceae? (please check, I m note sure)

Thermomonosporae the correct name is Thermomonospora

The Streptosporangiales is an order of bacteria

Q3: Lines 323-324

Bacillota (synonym Firmicutes) is a phylum of bacteria NOT a genus

Burkholderiales is an order of Betaproteobacteria in the phylum Pseudomonadota NOT a genus

The Clostridia are a highly class of Bacillota (synonym Firmicutes)

Duganella is a genus in the Oxalobacteraceae family

Lachnospirales I think is an order please check

Haliangium is a genus of bacteria from the family of Kofleriaceae ? please check

Haliangiales is the order of the genus Haliangium

Caulobacteraceae is a family of a Phylum Proteobacteria

Caulobacterales is an order

Q4: Lines 325-326

Alphaprotobacter the correct is Alphaproteobacteria or α-proteobacteria is a class of bacteria in the phylum Pseudomonadota (formerly "Proteobacteria"), please check

The Myxococcota is a phylum of bacteria, please check

Polyangia, I think the correct name is Polyangium a genus in the family Microscaphidiidae

The Sphingomonadales is an order of the Alphaproteobacteria includes the family Sphingomonadaceae, please check

Nitrosomonadaceae, the correct name is Nitrosomonadaceae. Nitrosomonadaceae is a family of betaproteobacteria

Q5: Line 328

Gammaproteobacteria is a class of bacteria

Allorhombium etc are genus of the family Rhizobiaceae? Pleace check

Reply: LDA Effect Size analysis is an analytical tool for discovering and interpreting biomarkers (taxonomic units, pathways, genes) in high-dimensional data. It can enable comparisons between two or more groups, as well as comparisons between subgroups within each group, in order to identify species with significant differences in abundance between groups.

The classification corresponding to the names of bacteria and fungi has been checked and modified on line 329-347. The note of Figure 3 on line 323-324 has been explained

the first letter before each bacterial and fungal name represents: p_ represents phylum, o_ represents order, f_ represents family, g_ represents genus, s_ represents species.

Round 2

Reviewer 4 Report

Comments and Suggestions for Authors

Dear authors,

To my knowledge, Rhizoctonia oryzae (lines 576-578) is primarily recognized as a pathogenic fungus. In the research article https://doi.org/10.1016/j.jep.2021.113975, the authors concluded that five endophytic fungi—Aspergillus fumigatus, Aspergillus niger, Rhizoctonia oryzae, Rhizopus oryzae, and Syncephalastrum racemosum—were isolated. However, I do not see any direct evidence that Rhizoctonia oryzae is endophytic. Some Rhizoctonia-like fungi have been noted for their ability to form endophytic associations with higher plants, but this does not necessarily apply to Rhizoctonia oryzae. I have the same concern regarding Rhizopus oryzae, which is primarily recognized as a saprophytic fungus, meaning it typically decomposes dead organic matter. You can keep lines 579-578 as they are, or feel free to improve them.

Minimelanolocus species are known as saprobes, meaning they decompose dead organic matter. Some species are found on specific host plants, however, there is no explicit mention of these species being endophytic. It may be worth noting that this research article identifies Minimelanolocus for the first time as an endophytic organism. Please check.

Author Response

Comments and Suggestions for Authors

Dear authors,

To my knowledge, Rhizoctonia oryzae (lines 576-578) is primarily recognized as a pathogenic fungus. In the research article https://doi.org/10.1016/j.jep.2021.113975, the authors concluded that five endophytic fungi—Aspergillus fumigatus, Aspergillus niger, Rhizoctonia oryzae, Rhizopus oryzae, and Syncephalastrum racemosum—were isolated. However, I do not see any direct evidence that Rhizoctonia oryzae is endophytic. Some Rhizoctonia-like fungi have been noted for their ability to form endophytic associations with higher plants, but this does not necessarily apply to Rhizoctonia oryzae. I have the same concern regarding Rhizopus oryzae, which is primarily recognized as a saprophytic fungus, meaning it typically decomposes dead organic matter. You can keep lines 579-578 as they are, or feel free to improve them.

Minimelanolocus species are known as saprobes, meaning they decompose dead organic matter. Some species are found on specific host plants, however, there is no explicit mention of these species being endophytic. It may be worth noting that this research article identifies Minimelanolocus for the first time as an endophytic organism. Please check.

Reply: We deleted the original literature on line 576-578.

We rechecked the four endophytic bacteria and three endophytic fungi we obtained. We have found Sphingobium (reporting function) in reference [48], Tetracladium (reporting morphology) in reference [49]. We have added relative contents in line 576-581. Three endophytic bacteria (Acidibacter, Sphingomonas, and Variovorax) and two endophytic fungi (Cadophora and Minimelanolocus) were only reported that they were endophytes and were related to the plant growth.
